# Clinical Evolution of Preschool Picky Eater Children Receiving Oral Nutritional Supplementation during Six Months: A Prospective Controlled Clinical Trial

**DOI:** 10.3390/children10030495

**Published:** 2023-03-02

**Authors:** Carlos Alberto Nogueira-de-Almeida, Luiz Antonio Del Ciampo, Edson Zangiacomi Martinez, Andrea Aparecida Contini, Maria Eduarda Nogueira-de-Almeida, Ivan Savioli Ferraz, Matias Epifanio, Fabio da Veiga Ued

**Affiliations:** 1Medical Department, Federal University of São Carlos, Brazil—DMED UFSCAR Rod. Washington Luiz, km 235, São Carlos 13565-905, Brazil; 2Medical School, University of São Paulo, Brazil—FMRP-USPAv, Bandeirantes, 3.900, Ribeirao Preto 14049-900, Brazil; 3Nutrition School, University of São Paulo, Brazil—FMRP-USPAv, Bandeirantes, 3.900, Ribeirao Preto 14049-900, Brazil; 4Pediatrics Department, Pontifícia Universidade Católica do Rio Grande do Sul—PUC-RSAv, Ipiranga, 6681-Partenon, Porto Alegre 90619-900, Brazil

**Keywords:** food fussiness, dietary supplements, growth, nutrition rehabilitation, micronutrients, children

## Abstract

Objective: To determine if oral nutritional supplementation of picky eater children has a beneficial effect in addition to nutritional guidance on anthropometric parameters, nutrient intake, appetite, physical activity, and health complications. Methods: This is a randomized, single-blind, controlled clinical trial that included Brazilian picky eater children aged 24 to 60 months. The individuals were randomized into a control group (CG) (*n* = 17) and an intervention group (IG) (*n* = 18), and were followed up in seven meetings for 180 days (baseline plus one meeting every 30 days). The CG received nutritional guidance for food selectivity, while the IG received the same guidance plus oral nutritional supplementation. Anthropometric and nutrient intake assessments were carried out, and appetite, physical activity and health complications were investigated. Results: In the IG, the z-score of weight and height increased significantly over time (*p* < 0.05), while the body fat percentage (BFP) and BMI z-score remained unchanged. The percentage of inadequate intake of vitamins D, C and folate reduced in the IG over time compared to the CG (*p* < 0.05). In the IG, the score assigned by parents to the appetite scale increased over time (*p* < 0.05). There was no difference between the groups in the scores on the physical activity and global health scales, and in the number of health complications. Conclusions: Picky eater children that were supplemented increased their weight not by gaining fat, but due to an increase in stature, as shown by BMI z-score and BFP, that remained unchanged. Furthermore, they showed a decrease in inadequate micronutrient intake during the intervention. An improvement in appetite was also observed over time, attesting to the benefit of supplementation.

## 1. Introduction

Feeding difficulties (picky eating) are characterized as situations in which the child has less food acceptance than expected, with the possibility of causing physical, emotional and family disturbances, along with repercussions with regard to growth and development, depending on the duration, intensity, time of diagnosis, and the action of health professionals and family. It is estimated that about 62% of healthy infants and preschool children have some symptom of a feeding problem [1], which can either be a transitory situation or present throughout the growth period and persist into adulthood. The most common form of picky eating is selectivity [2], in which case the child is usually called a “picky eater”.

When picky eating leads to lower calorie, macro and micronutrient intakes, growth can naturally be expected to be impaired. In the United Kingdom, Wright et al. [3] found lower weight and height among picky eaters, with 11% of them below the fifth percentile of weight gain, a percentage three times higher than that found among children without picky eating. A previous study by the same group had already shown that many of the children diagnosed as having “failure to thrive”, at 15 months of age had previously presented some feeding difficulty [4]. Regarding micronutrients, picky eaters have variable outcomes [5]. Galloway et al. [6] found a low consumption of vitamins E and C, while Carruth et al. [7] observed a high prevalence of deficiency in calcium, zinc and vitamins D and E intakes.

The treatment of picky eating includes nutritional guidance measures, preferably with an interdisciplinary approach [8], and, in some cases, oral supplementation. Clinical experiences on the effect of food supplementation on the growth of children with low appetite and selectivity, or whose dietary pattern was monotonous and with the inadequate intake of micronutrients, demonstrated an improvement in the general state of health [9]. The use of supplements with adequate energy content and balance in the composition of vitamins and minerals has shown results both in maintaining nutritional status and in improving the situation of specific vitamin deficiencies, allowing the professional to guarantee the nutritional security of the patient, that is, growth and adequate development, while implementing nutritional reeducation techniques [10,11]. In general, studies carried out with supplementation use two daily doses of 200 mL of isocaloric profile products (1 calorie per mL), equivalent to an offer of 400 calories daily, with an adequate balance of macronutrients and complete in micronutrients, in addition to fiber [10,12]. In Brazil, this type of supplement has been termed a “complete supplement” as defined by the Brazilian Association of Nutrology [13].

However, the treatment of picky eating still presents some questions not completely answered by science. Despite being a multicausal problem, there is a bottleneck in the picture when considering the negative impacts on food consumption, and health consequences are verified in the short-, medium-, and long-term [14].

Although studies can be found that justify the prescription of a complete supplement for all children with picky eating, these exist in small numbers to date, not allowing the execution of systematic reviews and meta-analyses, for example. Therefore, it is reasonable to ask whether it would not be possible to obtain adequate results through nutritional guidelines alone, without supplementation. Thus, the present study seeks to contribute to these questions, comparing two groups of picky eating children, randomly constituted, treated with or without the use of the supplement. The main objective was to determine whether the supplementation of picky eating children has a beneficial effect, evaluating the evolution of weight, height, body mass index (BMI), body composition, macro and micronutrient intake, appetite, physical activity, and the health complications of children supplemented compared to non-supplemented.

## 2. Methods

### 2.1. Study Design

This is a randomized, single-blind, controlled clinical trial conducted from January 2021 to August 2022 in a private pediatric clinic in the city of Ribeirão Preto, Brazil. The study was approved by the Research Ethics Committee of the Federal University of São Carlos (UFSCar), process number 3.510.241. The clinical trial was registered in the Brazilian Registry of Clinical Trials, with Universal Trial Number (UTN): U1111-1223-7015 and register number RBR-6pxpvx. Written informed consent was obtained from the parents or guardians of all participants.

### 2.2. Eligibility of Participants

Medical professionals and nutritionists from the city of Ribeirão Preto, Brazil, were invited to refer patients who presented complaints, as experienced by the family, of feeding difficulties. All individuals referred to the private clinic where the study was conducted, from January 2021 to August 2022, were assessed according to the inclusion criteria, configuring a convenience sample. The inclusion criteria were age between 24 and 60 months and a BMI z-score between −2 and +2. Referred patients were considered eligible for the study if they were considered picky eaters [2]. A total of 46 children were referred, and 43 were considered eligible. Exclusion criteria were the presence of cow’s milk protein allergy, lactose intolerance, impossibility of oral feeding, neoplasia, renal failure, liver failure or heart disease undergoing to be treated, genetic syndrome, anorexia nervosa, autism, attention deficit hyperactivity disorder, chronic diarrhea or inflammatory bowel diseases, and growth retardation and development related to chronic diseases. Among the 43 eligible children, two were excluded due to these criteria. After inclusion in the study, the participants (*n* = 41) were randomly allocated into two groups: (1) a control group and (2) an intervention group. Six participants expressed a desire to interrupt their participation in the study. Figure 1 summarizes the participants’ selection flowchart (*n* = 35), which included 17 individuals in the control group and 18 individuals in the intervention group.

### 2.3. Randomization and Masking

All participants were randomised by a computer-generated list into one of the groups by simple randomisation at a ratio of 1:1. Allocation concealment was granted by Research Electronic Data Capture (REDCap). A pediatrician and a statistician were blinded to the study and were responsible for the randomization process, allocation concealment and outcome evaluation. A non-blinded nutritionist was responsible for the intervention. Participants were unblinded to the treatment proposed in the study.

### 2.4. Interventions and Control Groups

The control group received nutritional guidance regarding feeding difficulties in seven meetings over 180 days (baseline plus one meeting every 30 days). General guidelines for healthy eating in childhood were addressed, mixed with guidelines for picky eaters [15]. Nutritional guidance was provided on (1) the food groups of the Brazilian food pyramid; (2) the amount of food to be ingested at each meal; (3) the methods of introducing low-acceptance foods, combined with well-accepted foods, without using blackmail, fights or rewards; (4) the selection of food in the supermarket; (5) culinary activities to be carried out at home; (6) playful activities that stimulate visual, olfactory and tactile contact with food; (7) games involving the feeding of children’s characters (princesses, superheroes, etc.); (8) time in front of electronic equipment; (9) sleep time; and (10) feeding at school.

The intervention group received the same nutritional guidance from the control group in seven meetings over 180 days (baseline plus one meeting every 30 days). In addition, children received oral supplementation, with a product registered with the National Health Surveillance Agency (ANVISA) and available in the Brazilian market, with an energy density of 1 calorie per milliliter (mL). The supplement provided was a commercial formula (Milnutri Complete ^®^, Danone Nutricia São Paulo, Brazil), donated by the study sponsor. The dose of 200 mL was prescribed twice a day, daily, to all participants in this group, totaling 400 mL per day, which is equivalent to 400 calories per day, for 180 days. The preparation of the supplement was done by the parents at home.

### 2.5. Supplement Composition

The detailed nutritional composition of the supplement offered to the intervention group is described in Table 1.

### 2.6. Study Outcomes and Dynamics of Intervention

The primary outcome measures were anthropometry and body composition analysis (bioimpedanciometry). As secondary outcomes, we recorded nutrient intake over time. As a tertiary outcome, we recorded the score on the appetite scale, physical activity scale, global health scale, and health complications over time. The moments in which these outcomes were evaluated are described in Figure 2. There were no protocol changes after the start of the study.

#### 2.6.1. Anthropometry

Participants were weighed on a Welmy^®^ brand (Santa Barbara do Oeste, Brazil) digital scale while only wearing underwear. Height was measured in a Seca-type wall estadiometer, with the patient standing barefoot, with their body straight and the nape touching the wall. Measurements were made according to standardized methodology [16]. After weight and height measurement, the body mass index (BMI) was calculated. Weight, height and BMI values were also obtained in z-score, according to World Health Organization (WHO) curves, in all consultations. Due to the physical growth characteristic of the pediatric age group, the WHO recommends that all reporting on weight, height and BMI be done using z scores, which is why this was our option in the present study [16].

#### 2.6.2. Bioimpedanciometry

The percentage of fat was measured using the bioimpedanciometry equipment *InBody*^®^, model 270 (Rio de Janeiro, Brazil), which includes equations appropriate for the age group involved. Before the examination, the patients fasted for 4 h and, during the execution of the test, they were only wearing underwear. The body fat percentage (BFP) was obtained at moments T0, T3 and T6.

#### 2.6.3. Assessment of Nutrient Intake

One 24 h recall (R24h) was applied at moments T0, T3 and T6, and was obtained by the methodology of ‘multiple passages’ in three stages [17]. Estimated average requirement (EAR) and adequate intake (AI) of Dietary Reference Intake (DRI) were used to determine whether nutrient intake by the population was adequate [18]. Energy, carbohydrates, protein, lipids, minerals (iron, calcium, zinc and magnesium) and vitamins (vitamin A, D, C, B12 and folate) were determined. The diet data were double-checked during the transfer to Nutrilife Software (Nutrilife Nutrition Software, Maringá, Brazil), which was used to analyze food intake data.

#### 2.6.4. Appetite Scale

Since there are no validated appetite scales for pediatric patients in this age group in Brazil, appetite was evaluated using a qualitative scale, created by the authors for this study (not validated), in which the mother attributed a score from 1 to 10 for the following question: “Regarding your child appetite, write down the scale below the score that best corresponds to your perception, 1 meaning totally without appetite and 10 meaning a lot of appetite”.

#### 2.6.5. Physical Activity Scale

The aim of this analysis was to evaluate whether there was a change in the pattern of physical activity in a generic way, unrelated to sports activity, considering mobility, willingness to play, wakefulness, etc. Thus, it was evaluated through a qualitative scale, created by the authors for this study (not validated), in which the mother attributed a score from 1 to 10 for the following question: “Regarding your child disposition to play, move, walk, run, jump, write down the note that best corresponds to your perception, 1 meaning almost not moving and 10 moving a lot”.

#### 2.6.6. Global Health Scale

This was evaluated through a qualitative scale, created by the authors for this study (not validated), in which the mother attributed a score from 1 to 10 for the following question: “Regarding your general perception of your child’s health, write down the score below that best corresponds to your perception, 1 meaning not healthy and 10 meaning very healthy”.

#### 2.6.7. Record of Health Complications

Health complications during the intervention were collected using a form prepared by the authors in which parents wrote down the answers to the following questions: “During the last month, has your child had any health problems?”. When the answer was yes, the parents noted what problem(s) was/were observed. “Have you noticed any complications that you related to the supplement offered by the study?”. When the answer was yes, the parents described the observed problem(s).

Parents were asked to record these answers at home as they detected the problems, in order to avoid forgetting. Later, these data were discussed with the parents during the consultations, so that any doubts were resolved, and filling errors were corrected. The number of health complications that the participants had during all the consultations of the study was recorded in absolute numbers.

#### 2.6.8. Check on the Use of the Supplement (Intervention Group)

Parents in the intervention group received a form to fill in the total volume of supplement effectively consumed daily by their child. In follow-up meetings, parents were asked to return unused or partially used cans. The information in this form was compared with the total supplement returned to confirm its veracity. Any disagreements were discussed with the parents, and adjustments were made where necessary.

### 2.7. Statistical Analysis

Baseline data between the groups were compared by Student’s t test, Fisher’s exact test, and the Wilcoxon test. The means of anthropometric data (expressed in z-score), energy and nutrient intake (expressed in kcal, g, mg and μg/day) and supplement intake (expressed in mL) were compared throughout treatment by linear models of mixed effects including time × group interaction terms, fitted using the “lme4” package of the R program. When the interaction terms are significant, we have evidence that the effect of time on the outcome variable is different for different groups. The mean number of health complications and scores on the appetite, physical activity and global health scales were compared by Poisson models including random effects, which were also fitted using the “mle4” package of the R program. The validity of the regression models was verified by residual analysis. The significance level used was 5%. Statistical analysis was performed using the R program, version 4.1.1.

## 3. Results

### 3.1. Baseline Characteristics of Subjects

Thirty-five individuals participated in the study, 17 in the control group and 18 in the intervention group. There was no difference in mean age, gender, or anthropometric parameters between groups at the baseline (T0) (Table 2). Only the score assigned by parents to the appetite scale differed significantly (*p* < 0.01).

### 3.2. Energy and Nutrient Intake

There was no difference in energy and nutrient intake between the groups at baseline (T0). After starting supplementation (T3 and T6), the intervention group significantly increased the intake of iron, zinc and vitamins D, C, B12 and folate compared to the control group. The percentage of inadequate nutrient intake was similar among the groups at baseline (T0) and reduced in the intervention group at T3 and T6 for vitamins D, C and folate (Table 3).

When evaluating the intake of nutrients in the intervention group (without comparing them to the control group), there was a significant increase (*p* < 0.05) in energy, carbohydrate, iron and vitamins C, D, and B12 intake between moments T0 × T6 (data not shown in tables). In the control group, this increase in energy and nutrient intake was not observed over time, but there was a significant reduction (*p* < 0.05) in iron and vitamin C intake between T0 × T6 (data not shown in tables).

### 3.3. Changes in Growth Indicators over Time

The trajectories of BFP and weight for age, height for age, and BMI for age z-scores are shown in Figure 3. Baseline data showed no differences between groups. During the six visits (T1 to T6), all indicators remained statistically similar between the control and intervention groups. The participants of the control group showed no difference in the evolution of their z-score of weight, height, BMI and BFP (T0 × T6). In the intervention group, the z-score of weight and height increased significantly over time (T0 × T6) (*p* < 0.05), while the BFP and BMI z-score remained unchanged. The mixed linear model analysis results did not show significant time × group interaction effects when the z-score of weight (*p* = 0.422), height (*p* = 0.178), BMI (*p* = 0.648), and BFP (*p* = 0.496) were considered as dependent variables. Thus, we have no evidence that the effect of time on these response variables depends on the groups.

### 3.4. Health Complications and Scores on Appetite, Physical Activity, and Global Health Scales

The trajectories of the number of health complications and scores on the appetite, physical activity and overall health scales are shown in Figure 4. Baseline data showed no differences between groups, except in the appetite scale score. Over time, there was no difference between groups regarding the score on the scales evaluated (appetite and physical activity) and the number of health complications. In the control group, the number of health complications increased significantly between T0 and T6. In the intervention group, the score assigned by the parents for the appetite scale increased significantly between T0 and T6. There was no sample loss over time in relation to the baseline. The statistical model did not show significant time × group interaction effects when the appetite scale (*p* = 0.052), the physical activity scale (*p* = 0.356), the global health scale (*p* = 0.289), and the number of health complications (*p* = 0.950) were considered as dependent variables. Therefore, there is no evidence that the impact of time on these outcome variables varies depending on the group.

### 3.5. Supplement Intake

The adherence to supplement intake was 100% of the children belonging to the intervention group. The average volume (mL) ingested monthly (T1 to T6) is shown in Figure 5A. The mean percentage ingested was above 80% of the prescribed dose over time (Figure 5B), except in the last month (T6), which was 76.9% of the prescribed dose.

## 4. Discussion

Picky eating is a frequent condition that requires nutritional guidance and, often, supplementation. The present study aimed to compare two groups of children with picky eating who received clinical treatment over six months, one of which received oral nutritional supplementation. We observed that the supplemented children presented significant weight and height gain over time (T0 × T6) without raising their respective IMCs and BFP. This effect was not observed in the control group. Similar results were observed by Yackobovitch-Gavan et al. [9], who evaluated the effectiveness and safety of one year of nutritional supplementation on linear growth and weight gain in children between three and nine years of age. In this study, children that were supplemented and ingested at least 50% of the supplement offered (“good consumers”) showed significant height gain without concomitant BMI elevation. In another study [19], similar results were found among picky eater children between 24 and 48 months of age and who were at risk of malnutrition, supplemented with two types of supplements for 90 days; the two supplemented groups showed significant gains in the percentiles of weight, weight/height, and BMI in relation to the group not supplemented [19]. Huynh et al. [12] conducted a study to observe the impact of nutritional supplementation and dietary counseling among picky eater children between three and four years of age presenting nutritional risk. After 48 weeks, the authors found improvement in the weight/height, weight/age and height/age indices and, similarly to our study, did not observe excessive weight gain or obesity.

In the present study, despite the improvement in anthropometric indexes, there was no increase in BMI with the use of nutritional supplementation, which shows a proportional increase in the anthropometric parameters of these children. In a study conducted by Khanna et al. [19], an increase in BMI was observed after supplementation, but in this investigation the children initially had a lower weight percentile than that of height, which could help explain this finding. Thus, it is possible to observe that supplementation of eutrophic picky eater children, as we have shown, does not cause an undesirable increase in BMI.

A fundamental aspect that the data of the present study showed refers to the fact that supplemented children presented an increase in their height z scores, which did not occur in the control group. This increase reflects the fact that picky eater children may have their growth slowed as a result of their feeding difficulties, as demonstrated by Jung et al. [20], Chau [21], Taylor et al. [22], and Viljakainen et al. [23]. In a 2018 publication, Ghosh et al. [24] reached results very similar to the present study, showing that supplementation was able to promote the catch-up in 90 days of intervention, with an evident difference between the supplemented and the control groups. Khanna et al. [19], studying picky eater children at nutritional risk, were able to promote catch-up in the supplemented group. Huynh et al. [12] also observed an increase in height z-scores, but there was no control group for comparison.

To verify whether the weight gain observed among supplemented children is not actually the result of excessive fat deposition, it is essential that the evaluation contemplates the evolution of body composition, parallel to that of anthropometric measurements. Galloway et al. and Taylor et al., using dexacytometry, showed that picky eater children present lower BFP when compared to controls [22,25], but these researchers did not evaluate body composition changes after interventions. In our study, in the intervention group, there was no increase in BFP even with the occurrence of weight and height gain in this group. It is important to highlight that the assessment of BFP of supplemented patients reinforces the fact that supplementation allowed weight gain due to linear growth, ruling out that this growth was the result of weight gain due to the excessive accumulation of body fat.

Regarding food intake, studies show that picky eater children may have deficient micronutrient intake [26,27,28]. In our study, we found a high prevalence of inadequate nutrient intake in both groups, such as vitamins D, C, iron, and folate, in the baseline data. Carruth et al. [7] interviewed 118 mothers of picky eater children between 24 and 36 months of age, finding a high prevalence of calcium, zinc and vitamin D and E deficiency, along with extremely low dietary variety. Kutbi showed that picky eaters consume less vegetables and fruits, less protein, and higher amounts of trans fats [29]. The main characteristic of our study was the evaluation of how this nutrient intake behaved after nutritional guidance among the control and intervention groups, and to compare it after supplementation at 3 and 6 months (T3 and T6). In the follow-up of these individuals, in the intervention group, the intake of iron and vitamins C, D and B12 was higher over time (*p* < 0.05) when compared to baseline, showing that supplementation is able to minimize some of the nutrient deficiencies frequently found in picky eater children.

Some nutrients have a fundamental prominence in children’s growth, such as zinc and calcium, and their prolonged deficiencies can lead to anthropometric changes, as shown in the study published by Xue [28]. In our study, when we compared the intervention group at T0 and T6 after supplementation, a reduction of more than 50% in the inadequacy of calcium, iron, zinc and vitamins A, C and B12 intake was observed. The deficiency of various micronutrients may be related to anorexia, often present in this group, and, fundamentally, to the low dietary variety classically observed in selectivity. Zinc is an essential nutrient in several and numerous physiological functions, including immune and antioxidant functions, growth, and reproduction. There is evidence to suggest that zinc deficiency, besides impairing growth, may be closely involved with anorexia, if not as an initial cause, then as an accelerating or exacerbating factor [30].

The energy intake between the two groups was similar in the three times evaluated; however, in the intervention group there was a significant increase between T0 and T6, probably linked to the consumption of the supplement, giving a greater contribution of energy, vitamins and minerals. The sum of higher nutrient intake may explain the improvement observed in the z-score parameters of weight and height in this group.

Regarding the change in appetite, guidelines and/or nutritional interventions used in isolation to improve food refusal have shown conflicting results. In the present study, children in the control group received dietary guidance, but showed no improvement in the appetite scale score. On the other hand, children in the intervention group showed a significant increase in the score of the appetite scale, showing an improvement in food intake according to the parents’ perception. In the study by Khanna et al., [19] the researchers observed improvement in food acceptance in 63 individuals between zero and 21 years of age after a nutritional intervention program. Sharp et al. [31] also observed improved food acceptance in children between 13 and 72 months of age with chronic refusal of food and dependence on enteral feeding or oral supplementation in relation to the control group after a five-day nutritional intervention. In another study, Huynh et al. [12], using the “Visual Analogue Scales” scale, observed improved appetite among children between three and four years of age after dietary supplementation and dietary guidance. Naila et al. [32] observed improved appetite in children aged 12 to 18 months with severe malnutrition who received nutritional intervention for three months and psychosocial stimulation for six months. Using the “Early Childhood Appetite and Satiety Tool” scale to measure appetite in the children studied, the authors observed a significant improvement in the scale scores after the intervention; it is interesting to point out that the children in the control group (without malnutrition) who received dietary counseling but without nutritional supplementation and psychosocial support, also significantly improved their appetite scores after the observation period. In a study conducted in children between 17 and 32 months of age with severe malnutrition [33], there was an improvement in appetite after nutritional intervention performed for six weeks with a mixture of vitamins and minerals (“multivitamin + multimineral”); however, the same phenomenon was found in the placebo group. The correction of specific micronutrient deficiencies may contribute to increased appetite in children receiving nutritional interventions, while improved nutritional status and the consequent decrease in the frequency of infectious episodes may help explain increased appetite, even in children who have not received supplementation. However, it should be emphasized that different ways of evaluating appetite improvement after nutritional interventions can also sometimes contribute to divergent results.

Several studies have shown the beneficial effects of nutritional interventions on physical activity and the quality of life of children and adolescents. Huynh et al. showed that, after nutritional supplementation and dietary counseling, children aged three to four years at nutritional risk showed increased physical activity, measured by the “Visual Analogue Scales”; the authors even considered this as a possible factor that may explain the increased appetite in picky-eater children receiving supplementation [12]. Verjans-Janssen et al. [34] studied the effect of nutritional interventions (without the use of supplements) and stimulation in children aged four to 12 years in the school environment and observed an increase in physical activity. Yu et al. [35] observed, among other benefits, decreased social anxiety (defined by a scale—Social Anxiety Scale for Children—SASC) after a program of nutritional intervention and stimulation of physical activity of eight months in Chinese children between eight and 11 years of age. In the present study, no differences were found between the level of physical activity and the general perception of health between the groups or within the groups throughout the study. An intervention that was more focused on nutrition (and not physical activity) may help explain these data. Moreover, the fact that we studied healthy children (even picky eaters) may have hindered parents’ perception of changes in the general health status of their children. In addition, the instrument used in our work regarding the general perception of children’s health by their parents may have been something “nonspecific”, as opposed to studies aiming to detect “punctual” changes, such as anxiety.

Some studies on nutritional supplementation have shown varied results in reducing the incidence of disease. Fisberg et al. [10], in 2002, found a reduction in the number of days of disease among picky eater children that were supplemented. Alarcon et al. [36] observed a reduction in episodes of upper airway infection in the group of picky eater children between three and five years of age after nutritional supplementation and dietary counseling when compared to the group that received only dietary guidance, but the same effects were not found in relation to gastrointestinal symptoms. In the study by Huynh et al. [12], a reduction in the number of days in which children had acute diseases was observed throughout the study, especially regarding diarrheal episodes and respiratory infections. In the study of Dossa et al. [33], involving children between 17 and 32 months of age with severe malnutrition, there was a significant decrease in diarrhea episodes in the supplemented group after nutritional intervention; however, the same effect was found in children in the placebo group. Ghosh et al. [24] observed a reduction in respiratory infection episodes in Indian picky eater children between two and six years of age after nutritional intervention and dietary counseling when compared to those who received only food guidance. However, it is also known that the incidence of infectious episodes in these studies is often based on information collected retrospectively, which can lead to memory impairment, often compromising the analysis of the results obtained and generating controversial findings.

In our study we did not observe differences in the number of health complications in the intervention group over time; however, in the control group there was a significant increase in such episodes. Due to the importance of micronutrients in the proper functioning of the immune system, combined with the fact that picky eater children are at risk of deficiency of these nutrients, this may help explain the decrease in infectious episodes after nutritional supplementation in the previously mentioned studies that enrolled picky eater children with compromised nutrition status. In our study, we studied eutrophic children, and the presence of illness is unlikely, but it is intrigant that non- supplemented children exhibit more disease episodes than supplemented. It is known that, for epidemiological reasons, many childhood diseases have annual periods of higher incidence, and it is possible to speculate that the present study has gone through some of these periods, and supplementation may have protected the intervention group, unlike the control group that presented several episodes of disease during the study period.

In the present study, it was observed that 100% of the children ingested the nutritional supplement, with consumption of more than 80% of the recommended amount, except in the last month (T6), when consumption was 76.9%. Khanna et al. [19] obtained high adherence (99%) to the consumption of two types of supplements offered for 90 days to two different groups of picky eater children between 24 and 48 months of age at risk of malnutrition; in this study, a good compliant to supplementation was considered as the intake of at least 75% of the recommended volume. Ghosh et al. [24], studying the impact of nutritional intervention and dietary counseling performed in children between two and six years of age, observed high adherence to supplementation among the children studied (98.4%); furthermore, in this study a good adherence to supplementation was considered as the intake of at least 75% of the recommended volume. In our study, in addition to the notes made by parents regarding the amount of supplement consumed by children, there was a request for the return of the supplement not consumed or partially consumed, which allowed for the findings to be more reliable. Huynh et al. [12] obtained 100% adherence to the use of the prescribed supplement, and 85% of the children used both daily doses. The authors also mention that the use of oral supplements (instead of use via nasogastric tube) may have facilitated the ingestion by the children studied. Wright et al. [3] showed that 2.5-year-old picky eater children have a marked preference for liquids (especially dairy and similar), which are easier to accept, and this fact may help to explain the high adherence of children to oral supplementation among several studies. It is possible that, in our study, the sweet taste of the supplement helped with the acceptance.

The present study has some limitations. Because of the COVID-19 pandemic, enrollment of participants was problematic, and the study involved a smaller sample size than expected. In addition, we used simple randomization in a small group. However, despite this problem, both groups were balanced at baseline values. Furthermore, despite the small sample size limiting the significance of the results, the statistical analysis provided satisfactorily narrow confidence intervals for the means of the variables of interest, as we can visualize in Figure 3 and Figure 4. In addition, our results are in line with earlier published trials about oral nutritional supplementation in picky eater children. As with all research of this nature, the impossibility of direct supervision of the use of nutritional supplements by children can produce potential biases to the results. The relatively short intervention may have limited the observation of more significant changes in some anthropometric parameters less sensitive to “acute” changes in diet. Finally, the use of unvalidated scales to measure some outcomes, such as the appetite of the children studied, may have influenced the results obtained; moreover, as in any study that uses questionnaires, the memory of the interviewees may impact the findings.

## 5. Conclusions

Picky eater children that were supplemented did not gain excess fat. They increased their weight not by gaining fat, but due to an increase in stature, as shown by BMI z-score and BFP, that remained unchanged. Supplemented children had a higher intake of iron, zinc, folate and vitamins C, D, and B12 compared to controls. In addition, supplemented children showed improved appetite throughout the study. There was no change in parents’ perception of changes in physical activity, global health, or decrease in the frequency of health complications in supplemented children; however, regarding the latter outcome, these events were observed more frequently in children in the control group.

## Figures and Tables

**Figure 1 children-10-00495-f001:**
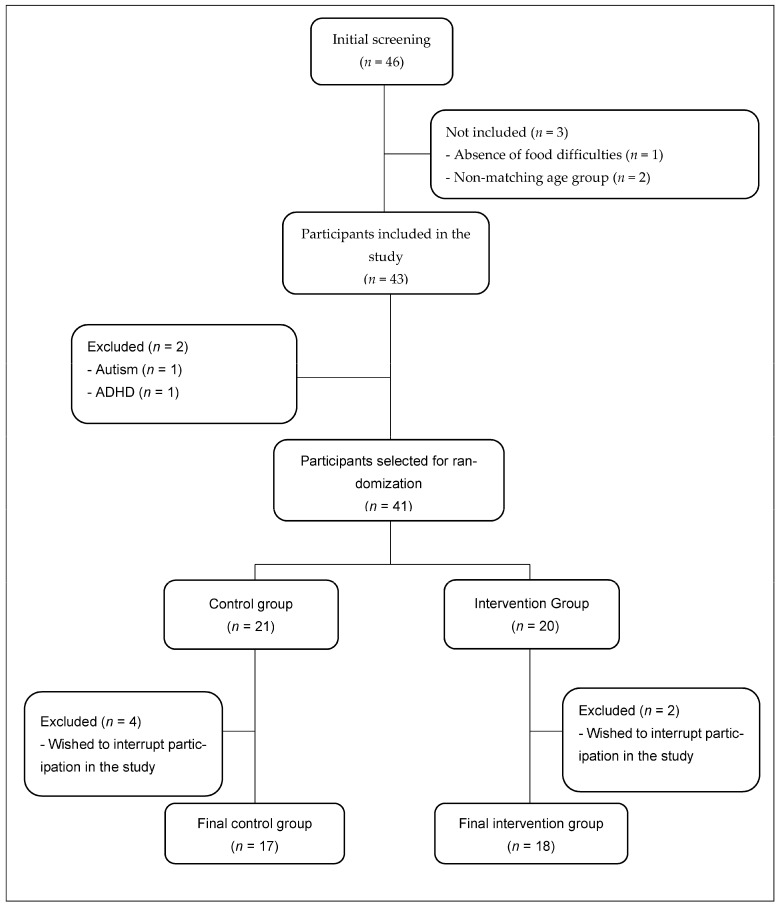
Participant selection flowchart.

**Figure 2 children-10-00495-f002:**
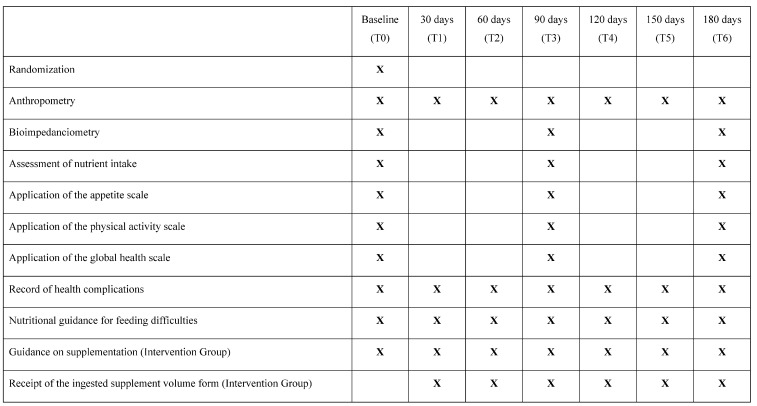
Details of the dynamics of the meetings during the 180 days of the study.

**Figure 3 children-10-00495-f003:**
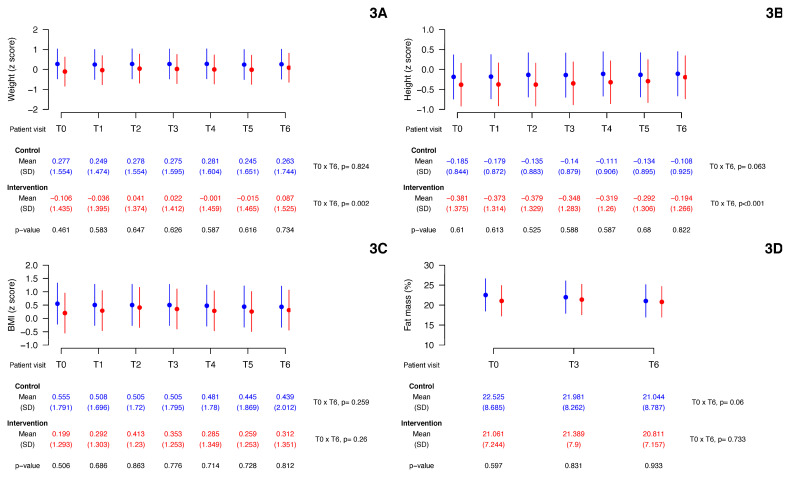
Changes in growth indicators over time. Vertical lines are 95% confidence intervals. *p*: longitudinal mixed effects linear model. (**3A**) weight z-scores vs. time; (**3B**) height z-scores vs. time; (**3C**) BMI z-scores vs. time; (**3D**) fat mass percentage vs. time.

**Figure 4 children-10-00495-f004:**
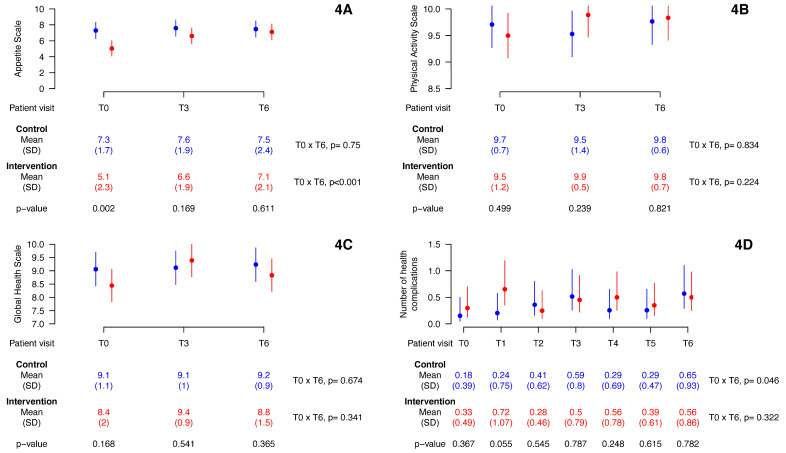
Health complications and scores on appetite, physical activity, and global health scales. Vertical lines are 95% confidence intervals. *p*: longitudinal mixed effects linear model. (**4A**) appetite scale vs. time; (**4B**) physical activity scale vs. time; (**4C**) global health scale vs. time; (**4D**) number of health complications vs. time.

**Figure 5 children-10-00495-f005:**
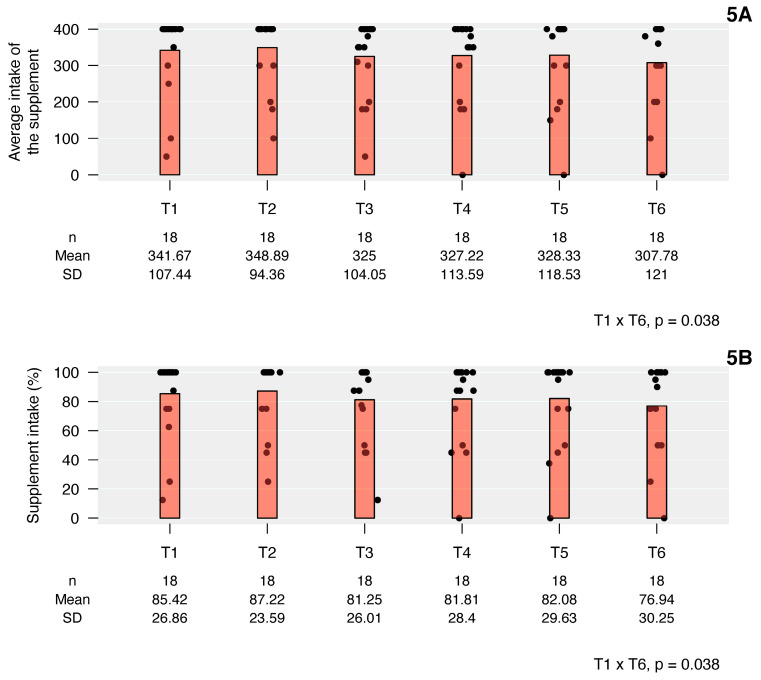
Intake (mL) and acceptance (%) of the supplement during the 6 months of study. *p*: longitudinal mixed effects linear model. (**5A**) average intake of the supplement vs. time; (**5B**) supplement intake (%) vs. time.

**Table 1 children-10-00495-t001:** Nutritional composition of the supplement used in the study (Milnutri Complete ^®^, Danone Nutricia São Paulo, Brazil).

Nutritional Information		
	Quantity per 100 g	Quantity per 100 mL
Energy value	434 kcal = 1817 kJ	100 kcal = 417 kJ
Carbohydrate	58 g	13 g
Sugar	28 g	6.4 g
Protein	13 g	3.0 g
Total fat	17 g	3.8 g
Monounsaturated fat	6.7 g	1.5 g
Polyunsaturated fat	3.1 g	0.7 g
Linoleic acid	2.6 g	0.6 g
Alpha-linolenic acid	0.5 g	0.1 g
Docohexaenoic acid (DHA)	65 mg	15 mg
Saturated fat	6.8 g	1.5 g
Trans fat	0 g	0 g
Cholesterol	14 g	7.0 g
Food fiber	4.4 g	1.0 g
Sodium	237 mg	54 mg
Folic acid	70 μg	16 μg
Pantothenic acid	1.1 mg	0.25 mg
Biotin	12 μg	2.7 μg
Choline	170 mg	39 mg
Niacin	2.7 mg	0.62 mg
Riboflavin	1.4 mg	0.31 mg
Thiamine	1.8 mg	0.40 mg
Vitamin A	350 μg-RE	80 μg-RE
Vitamin B12	2.2 μg	0.50 μg
Vitamin B6	3.5 ug	0.80 mg
Vitamin C	110 mg	25 mg
Vitamin D	12 μg	2.8 μg
Vitamin E	5.2 mg-α-TE	1.2 mg-α-TE
Vitamin K	25 μg	5.7 μg
Calcium	530 mg	121 mg
Chlorine	509 mg	116 mg
Copper	395 μg	90 μg
Chromium	14 μg	3.2 μg
Iron	6.7 mg	1.5 mg
Phosphorus	345 mg	79 mg
Iodine	49 μg	11 μg
Magnesium	46 mg	10 mg
Manganese	0.5 mg	0.12 mg
Molybdenum	7.7 μg	1.7 μg
Potassium	651 mg	148 mg
Selenium	18 μg	4.0 μg
Zinc	3.2 mg	0.74 mg
Taurine	32 mg	7.3 mg
Carnitine	13 mg	2.9 mg
Inositol	38 mg	8.6 mg
Osmolarity: 441 mOsm/l-Omega ratio 3:6:5.73 Caloric density: Proteins (12%), Carbohydrates (54%), Fats (34%)		

**Table 2 children-10-00495-t002:** Baseline data of the participants of the control and intervention groups.

	Control (*n* = 17) Mean (SD)	Intervention (*n* = 18) Mean (SD)	*p*-Value
Age (months)	47.0 (9.95)	42.33 (10.58)	0.19 ^a^
Boys/girls (n)	12/5	12/6	1.00 ^b^
Z weight score	0.28 (1.55)	−0.11 (1.44)	0.45 ^a^
Height z score	−0.19 (0.84)	−0.38 (1.38)	0.62 ^a^
BMI z score	0.56 (1.79)	0.20 (1.29)	0.50 ^a^
Fat mass (kg)	4.28 (3.13)	3.43 (1.6)	0.34 ^a^
Lean mass (kg)	5.88 (1.75)	4.82 (1.01)	0.05 ^a^
BFP	22.53 (8.68)	21.06 (7.24)	0.60 ^a^
Appetite scale	7.29 (1.72)	5.06 (2.31)	**<0.01 ^c^**
Physical activity scale	9.71 (0.69)	9.50 (1.20)	0.94 ^c^
Health scale	9.06 (1.09)	8.44 (1.98)	0.45 ^c^

SD: standard deviation; BFP: body fat percentage. ^a^ Student’s *t* test. ^b^ Fisher’s Exact Test. ^c^ Wilcoxon Test. Bold: *p* < 0.05.

**Table 3 children-10-00495-t003:** Energy and nutrients intake and percentage of inadequate intake at times T0, T3 and T6.

	T0				*p*-Value	T3				*p*-Value	T6				*p*-Value
	Control		Intervention			Control		Intervention			Control		Intervention		
Nutrient/Energy	Mean (SD)	Inadequacy (%)	Mean (SD)	Inadequacy (%)		Mean (SD)	Inadequacy (%)	Mean (SD)	Inadequacy (%)		Mean (SD)	Inadequacy (%)	Mean (SD)	Inadequacy (%)	
Energy (kal/day)	1371.8 (543.7)	-	1241.1 (363.6)	-	0.337	1449.2 (469.2)	-	1453.0 (402.1)	-	0.977	1338.5 (317.5)	-	1474.3 (236.3)	-	0.318
Carbohydrate (g/day)	182.1 (95.3)	23.5	167.4 (56.6)	38.9	0.464 ^a^ 0.332 ^b^	191.8 (41.7)	29.4	195.8 (59.5)	11.1	0.842 ^a^ 0.190 ^b^	185.1 (38.5)	11.8	199.3 (42.8)	16.7	0.479 ^a^ 0.680 ^b^
Protein (g/day)	50.3 (16.9)	5.9	45.7 (20.6)	11.1	0.535 ^a^ 0.587 ^b^	51.8 (24.5)	5.9	53.9 (24.4)	5.6	0.771 ^a^ 0.967 ^b^	48.1 (19.5)	0.0	54.3 (22.5)	5.6	0.397 ^a^ 0.995 ^b^
Lipid (g/day)	49.1 (17.31)	35.3	43.2 (16.9)	72.2	0.382 ^a^ 0.171 ^b^	52.7 (35.2)	41.2	50.4 (15.4)	16.7	0.736 ^a^ 0.062 ^b^	45.1 (16.3)	29.4	51.1 (11.3)	50.0	0.376 ^a^ 0.242 ^b^
Iron (mg/day)	10.9 (6.3)	11.8	8.2 (5.5)	27.8	0.087 ^a^ 0.249 ^b^	7.8 (4.3)	11.8	12.1 (4.3)	0.0	**0.008 ^a^** 0.991 ^b^	6.4 (2.9)	17.6	11.3 (3.7)	5.6	**0.002 ^a^** 0.285 ^b^
Calcium (mg/day)	959.8 (424.3)	41.2	760.4 (358.2)	38.9	0.163 ^a^ 0.890 ^b^	748.3 (514.5)	58.8	789.3 (272.6)	27.8	0.774 ^a^ 0.069 ^b^	909.7 (566.1)	47.1	892.6 (318.0)	16.7	0.904 ^a^ 0.061 ^b^
Zinc (mg/day)	6.4 (6.5)	35.3	5.2 (4.0)	38.9	0.451 ^a^ 0.826 ^b^	5.3 (4.1)	41.2	9.3 (7.1)	11.1	**0.017 ^a^** 0.055 ^b^	5.1 (2.8)	35.3	6.7 (3.0)	5.6	0.327 ^a^ 0.052 ^b^
Magnesium (mg/day)	135.9 (69.4)	35.3	106.2 (42.9)	27.8	0.149 ^a^ 0.633 ^b^	121.7 (58.1)	17.6	126.5 (57.7)	33.3	0.813 ^a^ 0.295 ^b^	121.9 (55.9)	35.3	125.6 (72.4)	16.7	0.854 ^a^ 0.216 ^b^
Vitamin A (mcg/day)	465.1 (312.6)	29.4	358.8 (312.3)	44.4	0.355 ^a^ 0.360 ^b^	419.6 (310.8)	23.5	425.2 (210.4)	11.1	0.921 ^a^ 0.339 ^b^	402.4 (232.2)	23.5	566.7 (392.9)	16.7	0.210 ^a^ 0.613 ^b^
Vitamin D (mcg/day)	5.5 (6.5)	82.4	4.9 (5.6)	83.3	0.742 ^a^ 0.939 ^b^	3.7 (3.7)	94.1	10.0 (4.1)	33.3	**0.001 ^a^** **0.002 ^b^**	4.6 (3.5)	94.1	10.5 (3.6)	27.8	**0.001 ^a^** **0.001 ^b^**
Vitamin C (mg/day)	83.2 (109.5)	35.3	48.2 (47.7)	22.2	0.137 ^a^ 0.395 ^b^	46.3 (78.2)	41.2	119.9 (63.2)	5.6	**0.002 ^a^** **0.030 ^b^**	43.8 (49.2)	41.2	119.6 (45.5)	5.6	**0.002 ^a^** **0.030 ^b^**
Vitamin B12 (mcg/day)	2.9 (1.9)	5.9	2.8 (1.4)	11.1	0.967 ^a^ 0.587 ^b^	3.5 (2.6)	23.5	4.1 (2.6)	0.0	0.349 ^a^ 0.587 ^b^	2.6 (1.9)	23.5	4.3 (2.2)	0.0	**0.023 ^a^** 0.587 ^b^
Folate (mcg/day)	92.0 (67.1)	76.5	84.0 (52.7)	77.8	0.652 ^a^ 0.927 ^b^	66.4 (34.1)	94.1	131.1 (62.5)	61.1	**0.001 ^a^** **0.042 ^b^**	67.1 (31.6)	94.1	115.3 (55.1)	61.1	**0.008 ^a^** **0.042 ^b^**

SD: standard deviation. ^a^ difference between groups regarding total intake (mean ± SD) at each moment (not a comparison over time)—longitudinal mixed effects linear model. ^b^ difference between groups regarding the percentage of inadequate intake (%) at each moment (not a comparison over time)—longitudinal mixed effects linear model. Bold: *p* < 0.05.

## Data Availability

Data supporting reported results can be requested.

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
