# Peer review of "Clinical Evolution of Preschool Picky Eater Children Receiving Oral Nutritional Supplementation during Six Months: A Prospective Controlled Clinical Trial"

_children, 2023, doi:10.3390/children10030495_

Round 1

Reviewer 1 Report

Dear Authors:

I read with great anticipation your manuscript looking at oral nutritional supplementation in picky eaters.

My overall question is - expecting a picky eater to not want to consume an oral nutritional supplement, how did the children rate the supplement in terms of palatability, taste, acceptance etc., and how were they motivated to consume a supplement? In other words what makes an oral nutritional supplement special that picky eaters accept it - sweetness (28 g of sugars)?

If the supplement provided is a commercial formula, then it needs to be named in the manuscript and if was or was not donated for the study, or if you paid for it. If you prepared the supplement in your own facility, then that needs to be clear in the manuscript.

These comments are in order of appearance in the manuscript:

Abstract: "...7 meetings for 180 days (1 meeting every 30 days)..." 7 x 30 = 210 not 180

Line 64: what is "nutritional security"? 

Line 67 justify that the supplement has an "adequate balance of nutrients".  How do you know which nutrients each picky eater needs without a baseline dietary analysis?

Section 2.4. Nutritional guidelines were very broad and general - are these specific for picky eaters? Numbers 1, 2, 4 and 8 seem to be not related to the population - please provide the reasoning behind the need to present guidelines on the 10 listed topics to picky eaters and their caregivers. 

Table 1: spaces between numbers and their units, the nutrient listed after biotin is "Hill" - type-o? use mcg and ug for micro, please use the SI symbol Mu to denote micro. For vitamin E use the alpha symbol or spell it out. "omega ratio" is that n-3:n-6 or n-6:n-3? Macronutrients are usually not plural.

Section 2.6.1: why fast for 4 hours before bioimpedence - would this affect hydration status?

Section 2.6.5: Is the physical activity scale validated - more details needed.

Line 213 "k" for kilo and micro symbol

Section 3: It seems that all the data is not provided. It would help greatly to review the study if all assessments were provided. Growth chart percentiles are missing, Absolute fat mass and weight etc are also missing - only z scores. The footnotes (table 3) for the p-values are confusing - why is it not based on time? Explain thoroughly the p-values. If the appetite score was lower in the picky eaters why did they consume the supplement, does it increase appetite?

Figure 3: if weight increased but body fat % remained the same, then for example 30% body fat for a 90 kg person = 30 kg fat mass, if they increased body weight to 100 kg then 30% body fat = 33.33 kg fat mass. It is more body fat.

In addition, if height increased in the intervention group, then the BMI calculation changes - if you divide by a larger number the BMI will be lower.

Actual numbers for all measures should be reported in addition to z-scores.

Appetite scale needs more explanation. How is number of health complications a decimal - is it divided by the number of participants - if yes please state how you got to the numbers.

Line 295-296: if 50% of the supplement is sufficient to get a gain in height, then should the recommendation be one supplement per day?

Line 305: "did not observe weight gain and obesity" - again if height increased the BMI will decrease - this is why absolute numbers make more sense. BMI does not seem to be useful in this case.

Line 312-313: you should see an increase in BMI if the children are increasing calories and weight.

Line 358-359: are you talking about anorexia or Anorexia Nervosa?

Author Response

Dear Reviewer,

Thanks for the comments and suggestions. The authors' peer-to-peer responses are described below and all changes we have made are highlighted in the text.

Point 1: My overall question is - expecting a picky eater to not want to consume an oral nutritional supplement, how did the children rate the supplement in terms of palatability, taste, acceptance etc., and how were they motivated to consume a supplement? In other words what makes an oral nutritional supplement special that picky eaters accept it - sweetness (28 g of sugars)?

Some data from the literature show that picky-eater children have a marked preference for liquids, which facilitates supplement adhering. In fact, other studies cited in our article show that other researchers also observed adequate acceptance of the supplement. At the same time, it is a fact that acceptance was not 100%, according to figure 5B, but the consumption was good enough to impact some health aspects. Please look at the following text that is included at the discussion: “Wright and col. showed that 2.5-year-old picky eater children have a marked preference for liquids (especially dairy and similar), which are easier to accept, and this fact may help to explain the high adherence of children to oral supplementation among several studies.”

We also included a new text: “It is possible that in our study the sweet taste of the supplement helped with acceptance”

Point 2: If the supplement provided is a commercial formula, then it needs to be named in the manuscript and if was or was not donated for the study, or if you paid for it. If you prepared the supplement in your own facility, then that needs to be clear in the manuscript.

Response 2: Thanks for the suggestion. We used a commercial formula donated by the study sponsor. The preparation of the supplement was done by the parents at home. We added this information to the text.

Point 3: Abstract: "...7 meetings for 180 days (1 meeting every 30 days)..." 7 x 30 = 210 not 180.

Response 3: The 7 meetings refer to the "baseline plus 6 meetings", that is, "baseline plus 1 meeting every 30 days, totalling 180 days of study". We write this information in more detail in the text.

Point 4: Line 64: what is "nutritional security"?

Response 4: We refer to proper growth and development. We add this information to the text.

Point 5: Line 67: justify that the supplement has an "adequate balance of nutrients".  How do you know which nutrients each picky eater needs without a baseline dietary analysis?

Response 5: We agree that a particular picky eater may have a deficiency in protein intake, while another has a deficiency in lipid intake, for example, and this requires an individualized analysis of food consumption. However, the aim of the study was not to supplement only the deficient nutrient in the picky eater's diet, but all nutrients, based on the premise that most nutrients were ingested in a deficient manner by these children. In this case, a complete nutritional supplement must be used, whose proportion of nutrients is balanced and adequate according to the nutritional needs of the age group, according to the DRIs.

Point 6: Section 2.4. Nutritional guidelines were very broad and general - are these specific for picky eaters? Numbers 1, 2, 4 and 8 seem to be not related to the population - please provide the reasoning behind the need to present guidelines on the 10 listed topics to picky eaters and their caregivers.

Response 6: We used general guidelines for healthy eating in childhood, mixed with guidelines for picky eaters. We add this information to the text. Step 1 aims to show parents the diversity of foods available in Brazil and the proportions in which the food groups should be consumed daily, since many parents do not know how to diversify their children's diet. Step 2 aims to teach parents that they should serve small amounts of food at meals (instead of forcing the child to eat the entire amount on the plate, which parents think is ideal) - this is a recommendation for picky eaters (Kerzner, 2009, Clinical Pediatrics). Step 4 aims to show the child the existence of different foods, knowing and getting used to different colors, textures and smells, especially when parents take them to the section where they sell vegetables, fruits, greens, and allow the child to choose an option to take home, where she will then perform step 5. Step 8 aims to show the negative influence of distractions caused by electronic equipment, especially at mealtimes - this is a recommendation for picky eaters (Kerzner, 2009, Clinical Pediatrics).

Point 7: Table 1: spaces between numbers and their units, the nutrient listed after biotin is "Hill" - type-o? use mcg and ug for micro, please use the SI symbol Mu to denote micro. For vitamin E use the alpha symbol or spell it out. "omega ratio" is that n-3:n-6 or n-6:n-3? Macronutrients are usually not plural.

Response 7: Thank you! We have corrected all information as requested.

Point 8: Section 2.6.1: why fast for 4 hours before bioimpedance - would this affect hydration status?

Response 8: Intake of water and food, hours before the bioimpedance test, may interfere with the result.

Point 9: Section 2.6.5: Is the physical activity scale validated - more details needed.

Response 9: The physical activity scale is not validated. This information has been added to the text.

Point 10: Line 213: "k" for kilo and micro symbol

Response 10: Thank you! We have corrected all information as requested.

Point 11: Section 3: It seems that all the data is not provided. It would help greatly to review the study if all assessments were provided. Growth chart percentiles are missing, Absolute fat mass and weight etc are also missing - only z scores. The footnotes (table 3) for the p-values are confusing - why is it not based on time? Explain thoroughly the p-values. If the appetite score was lower in the picky eaters why did they consume the supplement, does it increase appetite?

Response 11: The footnotes in Table 3 have been rewritten to improve understanding. The p values refer to each moment separately. Time-based p-values were not presented in tables, but are described in the second paragraph of section 3.2

Regarding the appetite, other cited studies show that picky-eaters who are overfed tend to have increased appetite. This was very well documented in the study by Huynh (2015), which was cited in the discussion of this outcome.

Point 12: Figure 3: if weight increased but body fat % remained the same, then for example 30% body fat for a 90 kg person = 30 kg fat mass, if they increased body weight to 100 kg then 30% body fat = 33.33 kg fat mass. It is more body fat.

Response 12: We agree that there is an increase in the absolute numbers of body fat (kg) when weight and height increase, and when the percentage of fat is maintained. However, the data in Figure 3 only reflect the percentage of body fat (%). We believe that evaluating the percentage of body fat (%) is better than evaluating the absolute numbers of fat (kg), since there are children from 2 to 5 years old in the sample, and of different genders, which could indicate significant differences in the amount of absolute fat (kg), but which do not mean clinical alterations. Thus, international references that assess the amount of body fat are expressed in percentages, instead of kilograms (eg McCarthy, 2006).

Point 13: In addition, if height increased in the intervention group, then the BMI calculation changes - if you divide by a larger number the BMI will be lower.

Response 13: We agree. However, BMI did not significantly decrease in the intervention group, but remained unchanged. This is because there was also weight gain, in addition to height increase. And that is why the data are presented in z-scores (which are already corrected for sex and age), instead of absolute numbers of BMI (kg/m²).

Point 14: Actual numbers for all measures should be reported in addition to z-scores.

Response 14: Due to the physical growth characteristic of the pediatric age group, the WHO recommends that all reporting on weight, height and BMI be done using z scores, which is why this was our option in the present study. This information has been added to the text.

Point 15: Appetite scale needs more explanation. How is number of health complications a decimal - is it divided by the number of participants - if yes please state how you got to the numbers.

Response 15: We agree. In the appetite scale, we add the information that it was developed for this study, exclusively, and is not validated. It is a subjective evaluation of the parents, where they will score the child's appetite from 0 to 10, with 0 being a reduced appetite, and 10 a voracious appetite.

As for the number of health complications, the reasoning is correct. The number of health complications in the last 30 days was divided by the total number of participants in the group. For example, at T0, 3 health complications were reported in the control group. This total divided by 17 individuals indicates that there were 0.18 health complications per person in the control group at T0.

Point 16: Line 295-296: if 50% of the supplement is sufficient to get a gain in height, then should the recommendation be one supplement per day?

Response 16: According to the study by Yackobovitch-Gavan et al, yes. However, we did not find this result in our study, so we cannot confirm it in our results and discussion.

Point 17: Line 305: "did not observe weight gain and obesity" - again if height increased the BMI will decrease - this is why absolute numbers make more sense. BMI does not seem to be useful in this case.

Response 17: In the study by Huynh et al. weight gain was also verified, in addition to height gain, which prevents the reduction of the BMI value (either in absolute numbers or in z-score). But the weight gain was not excessive, so the authors concluded that the supplement does not lead to obesity. We believe that BMI is a good parameter to be evaluated. If there was only an increase in height, the BMI would actually decrease, either in absolute numbers or in z-score.

Point 18: Line 312-313: you should see an increase in BMI if the children are increasing calories and weight.

Response 18: We partially agree. In the present study, we found a significant increase in energy intake (line 248) and weight (line 265), but we did not find an increase in BMI because height also increased significantly. That is why we say that, with supplementation, there was a proportional increase in the anthropometric parameters of these children (line 320-321).

Point 19: Line 358-359: are you talking about anorexia or Anorexia Nervosa?

Response 19: Anorexia, i.e. loss of appetite

Reviewer 2 Report

Need to elaborate on the main objectives of the manuscript. Some data needs to be more convincing.  

Author Response

Dear Reviewer,

Thanks for the comments and suggestions. The authors' comments are described below and all changes we have made are highlighted in the text.

Point 1: Need to elaborate on the main objectives of the manuscript. Some data needs to be more convincing.

We have made important improvements to the texts, which can be seen in the newly submitted version. We tried to make the objectives clearer and improve the discussion, emphasizing statistical aspects that were not clear enough, in order to make the results more convincing. We also proofread the writing in the English language.

Reviewer 3 Report

Dear authors,

-According to the sentence in line 99 of the method section, the estimated sample size was 100 people, but in the end, 46 people were referred and only 35 people were analyzed. First, what was the basis for estimating the sample size and with what formula and method? Secondly, how was this problem of difference between 100 estimated sample size and 35 referred children solved?

-Is the method used for randomization is “simple randomization”?  Simple randomization results could be problematic in relatively small sample size clinical research, resulting in an unequal number of participants among groups.

- There is no explanation about the validity and reliability of the “Appetite scale”, “Physical activity scale” and “Global health scale”. During what steps was this tool designed and approved?"

-It is suggested to add the names of statistical tests in the footnotes of other tables, similar to Table 2.

-Time*groups interaction analysis was not conducted for “Energy and nutrient intake”? In other outcomes that have been investigated, its meaning has not been interpreted! What is your interpretation for lack of statistical significance of time *group interaction? In my opinion, comparing one group alone over time without considering the other group is not valuable, and if interaction analysis is done, more attention should be paid to its results.

-In section “3.5. Supplement intake”, If I understood correctly, please specify that this results are for intervention groups only.  

Regards

Author Response

Dear reviewer

Thank you very much for your suggestions. Please find below our answers.

-According to the sentence in line 99 of the method section, the estimated sample size was 100 people, but in the end, 46 people were referred and only 35 people were analyzed. First, what was the basis for estimating the sample size and with what formula and method? Secondly, how was this problem of difference between 100 estimated sample size and 35 referred children solved?

  • In fact, the estimated number would be 100 children at the beginning of the study, but due to the COVID-19 pandemic, enrollment of participants was very problematic. Anyway, even before the pandemic, we would intend to use a convenience sample. We describe this problem in the limitations of the study.

-Is the method used for randomization is “simple randomization”?  Simple randomization results could be problematic in relatively small sample size clinical research, resulting in an unequal number of participants among groups.

  • You are correct. Simple randomization can prove problematic in small samples, but both groups (control and intervention) are similar in baseline values/characteristics. We acknowledge this as a limitation of the study in the text (last paragraph of the discussion section).

- There is no explanation about the validity and reliability of the “Appetite scale”, “Physical activity scale” and “Global health scale”. During what steps was this tool designed and approved?"

  • As written on line nº 180, there are no validated Brazilian scales for these variables. We acknowledge this as a limitation of the study in the text (last paragraph of the discussion section).

-It is suggested to add the names of statistical tests in the footnotes of other tables, similar to Table 2.

  • Done.

-Time*groups interaction analysis was not conducted for “Energy and nutrient intake”? In other outcomes that have been investigated, its meaning has not been interpreted! What is your interpretation for lack of statistical significance of time *group interaction? In my opinion, comparing one group alone over time without considering the other group is not valuable, and if interaction analysis is done, more attention should be paid to its results.

  • Thanks for remark. In fact, we have done this analysis. To answer these questions based on this analysis we insert three excerpt in the text in sections 2.7, 3.3 and 3.4. There is no evidence that the impact of time on these outcome variables varies depending on the group.

-In section “3.5. Supplement intake”, If I understood correctly, please specify that this results are for intervention groups only.

  • Yes, you are correct. We added an excerpt that makes this information clearer.

Round 2

Reviewer 1 Report

Dear Authors:

Thank you for making revisions. Overall, an improvement, however, revealing the supplement was provided by a sponsor after the fact was not correct procedure. I have minor reservations about the findings and what they mean in the real world, and providing sugary supplements to children does not ultimately seem right. I suspect this study will be used to create evidence for the sponsors products, all data from the study should be reported so it can be read by others.

Regarding z-scores, it is okay in an academic manuscript, however, when you meet with mothers who are worried their child is not growing, z-scores will need some explaining. Weight for height and other growth chart measures have worked for many years, and they are easier to understand for mothers.